# RETRACTED: An Adaptive Hierarchical Network Model for Studying the Structure of Economic Network

**DOI:** 10.3390/e24050702

**Published:** 2022-05-16

**Authors:** Xiaoteng Yang, Zhenqiang Wu, Shumaila Javaid

**Affiliations:** 1Key Laboratory of Modern Teaching Technology, Ministry of Education, Xi’an 710062, China; yxt@snnu.edu.cn; 2School of Computer Science, Shaanxi Normal University, Xi’an 710062, China; 3Department of Control Science and Engineering, College of Electronics and Information Engineering, Tongji University, Shanghai 201804, China; shumaila@tongji.edu.cn; 4Frontiers Science Center for Intelligent Autonomous Systems, Shanghai 201210, China

**Keywords:** complex network, network dynamics, data science, network evolution

## Abstract

The interdependence of financial institutions is primarily responsible for creating a systemic hierarchy in the industry. In this paper, an Adaptive Hierarchical Network Model is proposed to study the problem of hierarchical relationships arising from different individuals in the economic domain. In the presented dynamically evolving network model, new directed edges are generated depending on the existing nodes and the hierarchical structures among the network, and these edges decay over time. When the preference of nodes in the network for higher ranks exceeds a certain threshold value, the equality state in the network becomes unstable and rank states emerge. Meanwhile, we select four real data sets for model evaluation and observe the resilience in the network hierarchy evolution and the differences formed by different patterns of hierarchy preference mechanisms, which help us better understand data science and network dynamics evolution.

## 1. Introduction

The financial and economic development of various regions differs depending on their historical development and geographical setting. The movement of people and the development of commodity trade have promoted economic finance among different regions [1]. In some cases, the economic and financial system is likely to have a certain lag in economic and financial development due to the different levels of different regions and inefficient allocation of resources [2,3]. At the beginning of 2020, the outbreak of the new crown epidemic had a certain devastating effect on the economic and financial development of different countries and regions. Even in some areas, the economy and finance were in a “pause” phase, significantly affecting people’s lives [2]. The data from different countries, regions, and cities reflect different degrees of impact [4,5,6]. The complex network is used to depict the problems shown by the data promptly and construct different levels of differential structures to avoid economic and financial losses to the maximum extent possible [7]. The extreme network structure in each industry commodity is the only supplier of goods to other industry commodities. Ozsoylev et al. [8] consider the timing of trading in financial investments and foreign exchange markets as synchronous, describing the importance of investors grasp of important timing.

An important question is how hierarchical structures are formed in the economic and financial spheres and how they are stabilized through interactions between individuals [9]. Numerous studies have shown that the “winner effect” in human societies is also present in economic networks, i.e., people’s recognition of favorable activities increases their likelihood of winning in future activities. In human societies, winning a competition or battle leads to more support for individuals in future activities [10]. At the same time, the “winner effect” has a significant effect on the interaction between different financial institutions in economic networks.

### Literature Review

In the past few years, more and more scholars have started to study economic systems from the perspective of networks [4,8]. Describing an economic system as a network, the economic units in the system (e.g., individuals, firms, countries) are defined as nodes, and the interactions between different units are described as the connected edges between them. The trend of the network structure over time can provide information about the way the economic network system evolves. Hierarchical structures were first applied to study the social behavior of biological groups, which survive and develop through inter-hierarchical relations of superiority and inferiority [4]. At the same time, the economic and financial field demonstrates a robust hierarchical structure, where individuals from different levels of institutions play different roles in the economic network [5]. Researchers have used different methods to comparatively analyze the propagation of systematic financial risk under different scenarios and found a strong hierarchical structure. Garas [11] and Huang et al. [12,13,14] modeled the business cycle dynamics for systemic riskiness to assess the likelihood of failure of financial institutions at different levels and argue that the failure of a single entity in it triggers a series of failures in the system, i.e., the failure of one or more financial institutions leads to the propagation of systematic financial risk on a larger scale. Battiston et al. [15,16] introduce degree centrality in networks to compare different financial institutions and propose a new centrality measure DebtRank, which further extends the idea of centrality in networks, the impact of different levels of nodes on the network can be seen more clearly. Vodenska et al. [17,18,19] proposed a BankRank centrality metric based on the DebtRank idea and study the debt crisis providing evidence of the contagion of the 2007 financial crisis in equity and bond markets in emerging economies around the world.

Gai et al. [20] introduce the concentration and complexity of interbank structures into the hierarchical structure of the network, showing that different levels of financial structures increase the risk of the banking system when the network is subject to shocks. The changes in the structure of the world trade network over time are also analyzed. The study shows that as countries trade more and more closely with each other, there is an increasing heterogeneity in the choice of trading partners, so that it is very difficult to identify a representative country in the international trading system. Gale and Allen et al. [21] introduced an infectious disease model among viruses into the financial system network, treating financial structures with different levels of importance as different levels, and found that the propagation of financial risk depends on the inter-network different levels of inter-rank connectivity [22]. Moreover, a complete financial network structures are more stable than an incomplete network. In addition, network connectivity-based metrics explain stock market returns during financial crises: if the country in crisis is well integrated into the trade network, the crisis is more likely to spread; however, countries affected by a crisis shock that are well integrated into the network are, in turn, better able to eliminate the impact. When a financial crisis hits a specific part of the global trade network, the use of cascading and propagation issues in the network can help explain and understand the process of financial crisis propagation. Langfield and Fricke et al. [23] used a maximum entropy estimation method to compare the riskiness of financial networks. Cont [24] introduces a “contagion index” to measure the importance of financial institutions, i.e., the higher the rank, the greater the “contagion index” and analyzes the risk of contagion rank in the network and applies it to the contagion effect of the global financial crisis during 1997–2012, suggesting that financial institutions can more easily improve risk sharing by diversifying shocks.

At the same time, the rapid growth of financial data is becoming more and more important to reflect the connection between data through the web [25]. Different financial institutions are becoming more and more closely connected, and the financial structure has become dynamically diverse. Page et al. [18] propose the classical PageRank algorithm to measure people’s interest in different web pages, which can help us better understand people’s attention to different financial institutions. Applying it to the financial system, the attention of different individual units can be better identified. De et al. [26] propose the SpringRank algorithm to infer the hierarchical structure of different nodes in the network based on the physical model. Extending their study to economic networks so that the variability of individual interactions of different units in economic networks can be better studied. Hickey et al. [27] study the hierarchical structure in social networks and investigate the stability of networks as well as the phenomenon of clustering, which can be applied to different hierarchical financial networks to ensure that the networks do not get out of control. Sayama et al. [28] propose a two-layer temporal network model to enable a better understanding of the evolutionary nature of networks. Lu et al. [29,30,31] study the key nodes in heterogeneous networks and apply their application to different disciplines, providing a theoretical basis for identifying key nodes in financial networks [32,33].

Based on the above literature, the current research needs to address the following questions: (1) The phenomenon of ranking in economic networks is seen everywhere, while there is very little research on what factors lead to the emergence of ranking in networks. (2) Whether the rank differences exhibited by the same economic network dataset are consistent across node rankings, and also whether the most appropriate node scoring function exists for different economic network data. (3) In the economic network node interaction, what is the hierarchical position of the nodes that really generate the association in the network. Therefore, in order to better explore the network structure of economic finance and to solve the problem of shaping and sustaining different hierarchical relationships among networks. We have conducted the following work:In this paper, we propose a long-term effective network hierarchy evolution model. The model emphasizes the importance of information interactions among different financial institutions.This paper introduces parameters to control individual behavior and determines the hierarchy of nodes in the network through a function matrix.This paper proposes an egalitarian theory under long memory to determine the network elasticity and obtain the critical threshold of the system to ensure the hierarchical structure among networks.

The rest paper is structured as follows. The second part describes in detail the proposed network hierarchy evolution model as well as the systematic egalitarian theory; the third part verifies the correctness of the proposed theory through the simulation of real network data; the fourth part concludes the paper.

## 2. Materials and Methods

In economic networks, different institutions are more inclined to establish connections with highly visible or authoritative institutions. The interaction between institutions forms the theme of the network data, and the relationship between the data keeps changing over time. Competition and cooperation among financial institutions are key to their visibility, and the higher the visibility, the higher their value and the higher their relative rank in the network. Hickey et al. showed by examining hierarchical structures that dominance and prestige are two essential ways to form social status in social networks [27]. Similarly, hierarchical structures exist among financial institutions in economic and financial networks, which play a significant role in economic development.

### 2.1. Related Definitions

**Definition** **1.**
*(Degree k) For any given network, the degree k of a node is defined as the number of edges connected to it.*


**Definition** **2.**
*(Adjacency matrix 
A
) With a directed network 
G=V,E
, let 
aij
 be the case of connected edges from node i to j. If there exists 
i→j
, 
aij=1
; otherwise, 
aij=0
. Call 
A=(aij)
 as the network adjacency matrix.*


**Definition** **3.**
*(Diagonal matrix 
D
) let 
Din
 and 
Dout
 be diagonal matrices whose coefficients are the weighted-in and weighted-out degrees of the network, respectively 
Diiin=ΣjAij
, 
Diiout=ΣjAji
.*


**Definition** **4.**
*(Support ω) In an economic network, different units (e.g., individuals, firms, countries) interact with each other, and if there exists a directed edge 
i→j
 between two different units, we conceptualize the directed edge as support, i.e., j is supported by i.*


**Definition** **5.**
*(Support matrix 
ω(t)
) The interaction between nodes in the network changes over time, and the newly generated support relationship at moment t is defined as the support 
ω(t)
 in the network.*


### 2.2. Network Dynamic Evolution Model

The nodes in the network represent the individuals in the data and the connected edges illustrate the interactions between different individuals. In our proposed adaptive hierarchical network evolution model, nodes represent different economic and financial individuals, and directed edges represent interactions between different individuals. As time changes, new interaction information is generated between individuals, and new directed edges are generated based on the existing nodes in the network and the current hierarchy, after which these edges change with time. We represent the directed edges 
i→j
 as recognized support, i.e., the rank of individual *j* is higher than that of individual *i*. A directed weighted network represents the interaction information between *n* nodes, and the adjacency matrix 
A∈Rn∗n
 constitutes all the nodes in the network. 
aij
 is the weight of 
i→j
 in the network, representing the degree of support between two different nodes. The adjacency matrix 
A
 keeps changing with time according to the expression 
(1)
, where 
ω(t)
 is the support matrix representing the newly generated support relationship at the moment *t*. The “memory factor” 
φ∈0,1
 reflects the maintenance time of the support relationship, and the smaller m means that the expenditure relationship is more likely to be “forgotten”, based on which the proposed dynamic evolution model takes the general form of:
(1)
A(t+1)=φA(t)+(1−φ)ω(t),

where the new support relationship 
ω(t)
 depends on the ever support experience. The score *S* of *i* nodes is calculated by the score calculation function 
F:A→s
 in the related node ranking algorithm, and thus, the ranking order of each node is obtained. In the adjacency matrix in the directed weighted network, 
Din
 and 
Dout
 be diagonal matrices whose coefficients are the weighted-in and weighted-out degrees of the network. Next, we use three score functions:SpringRank: SpringRank is the latest research ranking algorithm by De et al. [34] The algorithm uses win-lose to quickly find the latent ranking in large networks, which has good adaptability to large systems, and most of the economic systems studied in this paper are large systems based on time series. Its treatment of the connections between network nodes as scalable physical springs is a rank recursive implication, where the rank of the supported individual is one rank higher than the rank of the supporting individual. Mathematically defined, SpringRank assigns a value to each node in the network, where the score 
s
 is a unique solution to a complex linear system [26].

(2)
Din+Dout−A+AT+βsIs=Din−Doute,

where the unit matrix 
I
 and the regularization parameter 
βs>0
 is a constraint added to the solution system to reduce the influence of noise on the system solution and to ensure the uniqueness of the resulting fraction 
s
, where e is the vector of ones.PageRank: PageRank is a classical ranking algorithm proposed by Page et al. [18], which is widely used in website promotion. In the economic system, if there is a connected edge between two units, that is, it is considered to produce support and there will be no other link interference information, which can well avoid the trouble caused by PageRank because it does not need to identify the connection characteristics, while at the same time, it can rank the importance of the system globally.Define score 
s
 as the PageRank vector of 
AT
, which is the the unique solution to the complex system

(3)
βpATDout−1+1−βpn−1eeTs=s

up to scalar multiplication. Standardize the score vector such that 
eTs=n
, where 
e
 is the unit vector. In the definition, the passing parameter 
βp∈[0,1]
 and we use the constant 
βp=0.85
 in our study.RootDegree: Its only considers the number of neighboring nodes supported and does not have transferability. The score 
s
 is the square root of the weighted number of support node *i*, i.e., 
si=Diiin
.

It can be seen that all three score functions can be viewed as rank ranking or centrality measures. However, unlike the SpringRank and PageRank scores, the RootDegree score is only related to neighboring nodes, and the impact of the score will not depend on the current hierarchy in which the individual is located.

After the score 
s
 is known, a new support relation 
ω
 exists in the network is obtained using a random utility model, a standard framework of discrete choice theory, which has recently been widely used in dynamic hierarchical models [4,12,19]. We consider a utility function of the form 
uij(s)=∑ℓ=1kρℓϕijℓ(s)
, where each 
ϕℓ
 is a smooth feature map; 
ρℓ
 is a preference parameter indicating relative importance of the *ℓ*th feature; and 
ϕijℓ(s)
 is the 
ij
th entry of 
ϕℓ(s)
. We use a special case with linear feature map 
ϕij1(s)=sj
, and quadratic feature map, 
ϕij2(s)=(si−sj)2
. To this point, a new support relationship is created

(4)
uij(s)=ρ1sj+ρ2si−sj2,

where we generally assume that 
ρ1>0
, 
ρ2<0
. The parameter 
ρ1
 represents the “prestige preference”, where a positive value of 
ρ1
 indicates a preference for a high scorer; 
ρ2
 represents the “proximity preference”, where a negative value of 
ρ2
 indicates a preference for an individual with a similar score to oneself. In addition, the probability of the final support relationship 
i→j
 can be expressed by the polynomial

(5)
pij(s)=euij(s)∑j=1neuij(s).
We obtain 
m∈N
 supports in the updated matrix 
ω
, where 
ωij
 gives the number of times *i* supports *j* within a time step. Expressions (1) and (5) represent the key features of our model. First, the dynamics in expression (1) indicate that the past support relationships are decaying at rate 
φ
. Second, expression (5) shows that the likelihood of a node receiving more support in a given time step depends on the probability of the distribution of support received earlier. For ease of understanding, the concept of inter-individual support is translated into the concept of rank in the network, i.e., the more support an individual receives, the higher the rank it will be in the network.

Figure 1 is a schematic diagram of the dynamics of our model, reflecting the relative change in rank at different nodes over a time horizon. The horizontal axis *t* represents the moment change, and the vertical axis *s* represents the rank score; the higher the score rank will be on the vertical axis. The dashed line 
φ
 represents the new support relationship at time *t*, and the solid line represents the pre-existing support relationship in the system. The lighter the color, the less important the relationship is and the more likely it is to be “forgotten”. At *t* = 1, the model is initialized and the support received by different individuals is recorded in the network adjacency matrix 
A
. The scoring function takes the adjacency matrix 
A
 as input and the score vector *s* as output. A new support relation 
ω
 is then obtained according to expression (5), and this new relationship is weighted by 
1−φ
 and combined with the previous support relation weighted by 
φ
. As time changes, this process is repeated, and the new support relations in the network gradually replace the old support in the system and update the score vector *s*. It can be observed that most of the support is a “short” leap, which is a pervasive pattern in most network data.

To facilitate the observation of the behavior among nodes in the network, the concept of “rank vector” 
γ
 is introduced, whose *j*th element 
γj=n−1∑ipij
 represents the possibility of new support to the *j*th node. If all the 
γj
 in the network are equal, then the system state is balanced at this point, otherwise there will be differences.

Figure 2 shows the rank variation of the dynamics with different parameters when using the SpringRank score function. The left side of the figure shows the variation of rank vectors for different 
ρ1
 and 
ρ2
, and different colors indicate the rank of different nodes. The right side shows the adjacency matrix at *t* = 4000. It can be seen from subplots (c) and (g) that the hierarchical differentiation is more obvious in subplot (g). When 
ρ1
 is small, the system as a whole is approximately egalitarian, and for larger 
ρ1
, there is a more clear hierarchical structure. However, the network system is elastic, and later we will find the critical value of 
ρ1
 that brings a huge change to the network hierarchy evolution under different score functions. In addition, the impact of 
ρ2
 is more reflected in the stability of node ranking, as can be seen from Figure 2e,g that smaller 
ρ2
 can reduce the fluctuation brought by the rank evolution in the network, making the network state smoother.

### 2.3. Parameter Estimated

In order to be able to statistically infer the hierarchical structure in the network, for our model, a likelihood function is proposed that can support maximum likelihood parameter estimation and also allow direct comparison of different score functions. 
A(t)=A(t)t=0τ
 is used to represent the time series of the matrix at a fixed time *t*. With the maximum likelihood model, the parameters 
ρ
 are learned from the observed series of support matrices 
ω(t)
, 
ω(t)
 depending on the state sequence matrix 
A(t)
 just through the nearest state 
A(τ)
. Thus, we can decompose the observed probability of a set of parameters to be determined as

(6)
P({ω(t)};A(0),φ,ρ)=∏t=0τP(ω(t);A(t),ρ).


The expression (6) is an implicit function and the right-hand side of the equation 
φ
 has vanished while 
ω(τ−1)
 and 
A(τ−1)
 depend on 
ω
. Let 
ki=Wi
, and 
Ki=eTki
. We get,

(7)
P(ω(t);A(τ),λ)=∏i=1nKi∏j=1nkij!∏j=1nγij(t)kij.
Integrating the terms of 
φ
 or 
ρ
 whose values do not depend as 
C(t)
 and then taking the logarithm of the expression, we get

(8)
logP(ω(t);A(t),ρ)=∑i=1n∑j=1nkij(t)logγij(t)+C(t).
The logarithmic probability of the entire sequence expression is

(9)
L(φ,ρ;{ω(t)},A(0))=logP({ω(t)};A(0),φ,ρ)=∑t=0τ∑i=1n∑j=1nkij(t)logγij(t)+C.


It can be seen that the dependence of 
ρ
 is expressed through 
γij
. Both 
φ^
 and 
ρ^
 are chosen as values for the parameter estimates:.

(10)
φ^,ρ^=argmaxL(φ,ρ;{ω(t)},A(0)).


The standard theory of maximum likelihood for exponential distributions shows that *L* is a convex function for 
ρ
 for any determined 
φ
. This suggests that 
ρ^
 can be solved by standard first or second-order optimization methods when 
φ^
 is known. let 
L*(φ;{ω(t)},A(0))
 be the optimized logarithm for a fixed 
φ
, and later optimize 
L*
 for 
φ
 to complete the maximum likelihood scheme. The global maximum is found by running multiple times using different initial values of 
φ
. Our model uses three parameters to fit thousands of observations, so overfitting is not a problem when training the data to evaluate the model.

### 2.4. Linear Stability

The behavior observed in Figure 1 suggests that there are mechanisms of different nature in the model under the 
ρ1
 “prestige preference”. When 
ρ1
 is small, the stronger institutions in the financial network do not exhibit a strong competitive advantage and the network as a whole is approximately egalitarian. However, for larger 
ρ1
, stronger individual institutions show a strong competitive advantage to limit the network to a stable hierarchical structure. We define a function 
f
 in the memory factor 
φ→1
 to characterize the critical value of 
ρ1
 for different scoring functions to bring about a large change in the overall network hierarchy evolution.

(11)
f(s,A)=limφ→1E[σ(φA+(1−ρ)ω)]−s1−φ,

where the expectation is with respect to 
φ
. If 
f(s,A)=0
 for all 
A
, then the score vector 
s
 is the key point of expectation in the model. Our choice of SpringRank, PageRank and RootDegree score functions allows us to derive conditions for the stability of grade evolution in the long memory limit.

We consider the eigenfunctions of the model, where 
ρl∈R
 denotes the relatively important feature parameter 
ϕl:Rn→Rn×n
 is the total feature mapping of the network and 
ϕijl(s)
 is the entropy of 
ϕl(s)
. Equation 
(4)
 is a special representation of the linear feature 
ϕij1(s)=sj
 and the quadratic feature 
ϕij2(s)=(si−sj)2
, while defining the network update rate matrix 
G=[n−1pij]
. Our goal is to obtain the stability of the system by first considering the Jacobi matrix of the rank vector 
γ
 in the system at a fixed point 
s0=θe
. Based on the previously defined “rank vector” 
γ=n−1GTe=n−1∑iγi
 and applying differentiation, we have:
(12)
∂γs0∂s=∑iΓi−γiγiT∑ℓ=1kρℓ∂ϕiℓ∂s,

where 
Γi
 and 
ϕi
 are the *i*th row of the *j*th feature mapping obtained by the network at 
s0
. When 
s0=θe
 and 
G=n−1E
, there is 
Γi−n−1I
 and 
γi=n−1e
. Thus, we have

(13)
∂γs0∂s=n−1I−n−1E∑i=1n∑ℓ=1kρℓ∂ϕiℓs0∂s≜Ms0;ρ


After that, this matrix is applied to our main results. As the time step *t* increases, we use 
δs=s(t+1)−s(t)
 and 
δA=A(t+1)−A(t)
 to denote the increment of 
s
 and 
A
 in the network.

SpringRank Scores

We develop the computation from the introduced properties of the score function. A SpringRank score vector 
s
 of a regularized 
β∈R
 matrix 
A
 is a solution of a linear system of equations

(14)
Di+Do−A+AT+βIs=di−do

where 
di=eTA,do=ATe
. Equation (Equation 2) is invertible when 
β>0
, i.e., 
e
 is the only solution to the equation. Therefore, assuming 
β>0
, define 
Lβ=Di+Do−A+AT+βI
 and 
Λ=Di−Do
. In the SpringRank function mapping, the vector 
s0=0
 is a fixed point of 
f
 and is the only mean fixed point in the dynamics. This fixed point point is linearly stable in the long memory limit if and only if the matrix 
M(0;ρ)−2n−1I−n−1E
 eigenvalue is strictly less than 
βnm
.

Starting from the analytic form of 
f
, the deterministic approximation 
f
 of the SpringRank vector is

(15)
f(s,A)=s+Lβ−1−βs−mn−1LGs−n−1e−γ

where 
LG=Γ+n−1I−G+GT
. We need to compute 
J(s0)
, the Jacobi matrix of 
f
 at 
s0=0
. The fixed point is stable when 
J(s0)
 has strictly negative eigenvalues. Calculating the derivative of (15).

(16)
∂f(s)∂s=I−Lβ−1βI+mn−1∂LGs∂s−∂γ∂s=I−Lβ−1βI+mn−1LG+Σ∂γ∂s−∂γ∂sST+eTsI−∂γ∂s


Evaluating this expression 
LG
 at 
s=0
 and 
G(0)=n−1E
 gives:
(17)
J(0)=−Lβ−1βI+mn−1LG−∂γ(0)∂s=−Lβ−1βI+mn−1I−n−1E2I−∑i=1n∑ℓ=1kρℓ∂ϕiℓs0∂s


Since 
Lβ
 is positive definite symmetric, 
Lβ−1
 is also symmetric. Therefore, the stability of the average immobile point in the SpringRank vector is determined by the eigenvalues of the matrix in parentheses. Multiplying by 
nm−1
, a sufficient condition to obtain the matrix is

(18)
I−n−1E2I−∑i=1n∑ℓ=1kρℓ∂ϕiℓs0∂s=M(0;ρ)−2n−1I−n−1E,

there are characteristic values not greater than 
βm
.

In particular, we have 
M(0;ρ)=ρn−1I−n−1E
 so require the matrix:
(19)
ρn−1I−n−1E−2n−1I−n−1E=n−1(ρ−2)I−n−1E

have eigenvalues smaller than 
βm
, so that the eigenvalues of the matrix can be calculated corresponding to the vector 
e
.

Then, any vector 
v⊥e
 is also an eigenvector with eigenvalue 
n−1(ρ−2)
. We therefore require 
n−1(β−2)<βm
, or 
ρ<2+βnm
 to complete the argument.

Therefore, in the SpringRank Linear model, 
s0=0
 is a linearly stable fixed point of 
f
 if and only if 
ρ<2+βnm
.

PageRank Scores: The PageRank score is the solution 
s
 of the linear system.


(20)
βATDo−1+(1−β)n−1Es=s,
where 
Do=diag(Ae)
. We directly use the PageRank model with linear features, scaling the parameter 
ρ
, assuming that 
s
 is normalized so that 
sTe=n
 and does not affect the analysis. Uniqueness is a direct consequence of normalization: if 
s=θe
, 
sTe=n
, then 
θ=1
.

Similarly, we next obtain a necessary condition describing the roots of 
f
. At any fixed point of 
f
, we have 
Do=mI
. Therefore it can be assumed that 
f
 is defined by 
s
.

(21)
βm−1nAT+(1−β)n−1Es=s


As the number of time steps *t* increases.

(22)
βm−1nAT+δAT+(1−β)n−1E(s+δs)=s+δs


(23)
βm−1nAT+(1−β)n−1Eδs+βm−1nδATs+o(1−φ)=δs,

where the term 
o(1−φ)
 includes the term involving the product 
(δAT)(δs)
 and relies on 
δs
 being a smoothing function of 
A
. In the case of long time *t*.

(24)
I−βm−1nAT−(1−β)n−1Eδs=βm−1nδATs


This expression gives an implicit representation of 
f
 via the relation 
f(s,A)=limφ→1E[δs]1−φ
. We can therefore enforce 
f(s,A)=0
 by setting 
E[δs]=0
, obtaining the necessary condition 
E[δAT]s=0
 for roots of 
f
.

(25)
0=EδATs=(1−φ)GT−ATs


Combining (21) and rearranging yields the nonlinear system.

(26)
GT+β−1(1−β)n−2Es=β−1n−1s
Then, the maximum eigenvalue of the matrix is 
β−1n−1
. Solving (26) by numerical iteration, followed by applying the standard eigenvalue solution 
s
 and updating 
G
 using the new value 
s
. To derive the linear stability criterion, we derive the derivative of 
s
 in (24) to obtain

(27)
I−βm−1nAT−(1−β)n−1EJ(s)=βm−1n∂∂s[GTs−ATs]=βm−1n∂∂sGTs−β−1mn−1s+β−1(1−β)mn−2Es=βm−1n∂∂sGTs−β−1mn−1s.
Then we evaluate at the linearly stable solution 
s0=e
, this becomes

(28)
I−βm−1nAT−(1−β)n−1EJs0=βm−1n−1E+βMs0;ρ−I.
When 
β<1
, 
I−βm−1n−1E−1=I+β(m−β)−1n−1E
 this matrix has a unique eigenvector 
e
 and eigenvalues 
1+β(m−β)−1
, while there are 
Ms0;ρ=ρn−1I−n−1E
 in the PageRank model, then we get:
(29)
Js0=βm−11+β(m−β)−1E+βρI+β(m−β)−1n−1EI−n−1E−I


The eigenvalues of 
J(s0)
 can now be obtained with the eigenvector 
e
 eigenvalue 
−1
. The eigenvalue of any vector orthogonal to 
e
 is 
βρ−1
 when and only when 
ρ<1β
. Thus, we obtain that in the PageRank-Linear model, the average root is linearly stable when and only when 
ρ<1β
.

RootDegree Scores

We first derive the functional form of 
f
:
(30)
E[s(t+1)∣A(t)]=E[A(t+1)∣A(t)]Te=φA(t)e+(1−φ)E[ω(t)]Te=φA(t)e+(1−φ)mn−1G(t)Te.
We then bring the expression into Equation 
(11)
, and 
n−1G(t)e=γ(t)
 to obtain:
(31)
f(s)=mn−1E[G]e−A(t)e=mγ−s.
We can determine that 
s0
 is indeed the unique average root of *f*. Assume that 
s=se
, and then that

(32)
f(s)=mγ(s)−s=mn−1−se.
When 
s=m/n
, the equation is equal to zero. We calculate the derivative

(33)
∂f(s)∂s=mM(s;ρ)−I.


This matrix has strictly negative eigenvalues, as long as the eigenvalue 
M(s0;ρ)
 is strictly less than 1/m. Next, the operation of the square root is considered as part of the identity to facilitate our understanding of the computation. Assuming that 
sj
 is the entry degree of node *j* and 
ϕj(s)=sj
, we get

(34)
Ms0;ρ=12n−1dρI−n−1E.


It can be seen that the eigenvalues of this matrix are still zero and related to the direction 
e
. For any direction 
v⊥e
, there exists an eigenvalue 
12n−1dρ
. We have:
(35)
1m>12n−1dρ.
and get

(36)
ρ<2dnm=2nm.


Thus, when and only when 
ρ<2nm
, 
s0=mne
 is a linearly stable immobile point of the function 
f
, we obtain the critical value of the giant change brought by the evolution of the rank when using the RootDegree scoring function.

Thus, we obtain the system critical value 
ρc
 by using the algebraic structure of the score function and the stability conditions for SpringRank, PageRank and RootDegree in the long-remembered time limit of averaging. An interesting phenomenon is that the “proximity preference” 
ρ2
 does not determine the initial state of the network hierarchy, only 
ρ1
 plays a role in the stability of the network hierarchy.

**Theorem** **1.***For the three scoring functions SpringRank, PageRank and Rootdegree, 
f
 has a unique linear stable root if and only if 
ρ1<ρ1c
, where*

ρ1c=2+βsnmSpringRank1/βpPageRank2nmRootDegree


Figure 3 illustrates the network rank stability prediction in the case of *n* = 8 nodes. To better explore the effect of 
ρ1
 on network stability, the average value over the last 1000 time steps is simulated by making 
ρ2=0
. The curves show the variation of the model under long memory. We divided the stability points into 2 groups, each with the same rank. For 
ρ1<ρ1c
, the ranks in the network are stable; conversely, at 
ρ1>ρ1c
, the network changes to unequal stable fixed points. Interestingly, in the PageRank and RootDegree models, there is a stable inequality state where a node receives support from almost all nodes (Figure 3a,b). It can also be seen that a network is in two stable (equality and inequality) equilibrium states, where nodes in both equality and inequality states are gaining support from other nodes, and which state the network eventually converges to depends on the initial conditions of the system. The SpringRank model shows different behavioral characteristics from the other two functions. At 
ρ1c
, the node ranks in the network are staggered, after which multiple high-ranking nodes become unstable as 
ρ1
 increases, until finally only very few high-ranking nodes remain. Although the system stability depends on the initial conditions of the network, it is likely that the system has more selectable stable states under this model. From this, it can be inferred that the SpringRank model is suitable for network systems with multiple different initial conditions and different rank states, which can be verified in the subsequent data.

## 3. Experiment Result

We compare models using four datasets to explore the network structure of financial institutions and address the issue of different hierarchical relationships between financial and economic networks. One is an airline network between different economic cities, one is an international trade network, one is an international investment network, and one is a network of economic capabilities among friends of universities. The data on airline networks between different economic cities are obtained from the 2016 “Open Airline Airport Database” provided by the openflights.org website (http://openflights.org/data.html (accessed on 10 March 2022)). The airline network consisting of economic cities and air routes reflects the intensity of urban connectivity, and the influence of a city can be reflected by its position in the airline network of major countries. The aviation network reflects the strength of connectivity between different economic cities and is an important path to study the structure of urban networks. In the study, we select the aviation data between the top 300 economic cities. The number of flights between different cities constitutes the adjacency matrix of the network, when city *i* has a flight to city *j* within time *t*, 
i→j
 produces interaction.

International trade data are derived from the Direction of Trade Statistics (DOTS) published by the International Monetary Fund, which contains trade volumes between individual countries and major trading partners from 2001 to 2016 [34]. International trade in this context refers to the cross-border exchange of economic organizations or governments with capital, goods, and services, etc. To avoid singularities in the data results, the 206 countries that conduct the most trade exports are selected for the study, and the network adjacency matrix is constructed with trade data. The 
i→j
 interaction is generated if country *i* and country *j* trade in time *t*. The international investment data are taken from the Coordinated Portfolio Investment Survey (CPIS) database provided by the International Monetary Fund. The database is a voluntary collection sponsored by the International Monetary Fund that collects data on portfolio investments, including equity transactions and debt securities, for individual countries and economies, and this paper uses data from 2001 to 2016. Again, to avoid singularities in the data results, the 206 countries or regions that conduct the most transactions are selected for the study, and the 
i→j
 interaction is generated if country *i* receives investments from country *j* within time *t*. The college student friend affordability data was accessed through the KONECT Network database [35]. One week after the start of a new semester, 17 fraternity members rank other brothers according to their financial ability and friendship level, where 1 denotes those who have similar financial ability to themselves and interact better with them; 16 denotes those who have a greater difference in financial ability to themselves and interact less well with them. When brother *i* ranks friend *j* in his top five at time *t*, it is considered that brother *i* regards member *j* as his good friend, i.e., 
i→j
 interaction occurs between brother members.

Next, the four network datasets are investigated using three score functions, SpringRank, PageRank, and RootDegree, respectively. Several key characteristics of the networks are reflected by parameter estimation, optimization to obtain log-likelihood values, and standard errors (Table 1). Similar behaviors were found for the different score functions in the four network datasets: 
ρ1>0
 and 
ρ2<0
. It can be shown that there is a general pattern in the time-dependent network hierarchy: the interaction between nodes does move toward higher levels, but will be more likely to support nodes that are not very different from their own levels rather than directly interacting with higher-level nodes directly, while the interaction between nodes will differ depending on the data set. In the economic airline network, because different cities have different economic levels, geographic locations, and airline transit capabilities, we find that the network exhibits distinct hierarchical characteristics, and cities with relatively low economies can improve their hierarchical position in the network by establishing links with high-ranking economic cities. In international trade networks and international investment networks, because the influence of economically strong countries is very important and trade services and investment are limited, it is relatively difficult for economically weaker countries to trade and invest, but the situation will improve with gradual economic development. Good friendships will be found in the network of economic capabilities in higher education where people with similar economic capabilities will build good friendships.

Different data will have different dependencies on different score functions, while the score functions will produce different features for different data, and each dataset is studied comparatively according to the differences of the models. The RootDegree model is preferred over SpringRank and PageRank in the economic urban airline network. Under the economic airline network dataset, the RootDegree score is a measure of the local city’s airline economy; the more airline routes a city has, the higher its score, independent of the prestige of the city where the local airline is located. The RootDegree score is consistent with previous research findings that the airline economy plays an important role in local production life, transportation, and air transport in the logistics sector [27]. In contrast, there is a strong dependence on SpringRank scores for international trade networks, international investment networks, and university economic capability networks, which suggests that transmissive prestige plays an important role in the structure of economic networks. In international trade networks and international investment networks, it is important not only to trade with other countries and make business investments, but also which countries to trade and make business investments with. This finding is consistent with the Hicket study [27,28], which suggests that the behavior of different regional interactions indicates their ability to make reasonable inferences about the location of hierarchical structures in the network. Similarly, building relationships with higher ranked classmates in a college friend’s affordability network may result in greater prestige than lower ranked classmates.

In addition to comparing different score functions, we also compare different models corresponding to the memory factor 
φ
 under the data set. As introduced earlier our model assumes that the effects of past support are decaying at a rate of 
φ
. 
t1/2=−log(2)/log(φ^)
 represents the half-life of the inferred dynamical system in terms of observation periods. When interpreting these estimated half-lives, the indirect effects of different individual interactions can far outweigh their direct effects. In the Eco avaiation data, the preferred RootDegree score half-life is 
t1/2≈4
 weeks. In both the international trade network and the international investment network, SpringRank scores have a lower half-life of 
t1/2≈2
 years, indicating that although only one-third of trade and investment transactions are directly “remembered” by the system after 4 years, these events affect 2 cycles of trade and investment events. The half-life of SpringRank scores in the College Friends econmic Capability Network is 
t1/2≈1.5
 weeks, indicating that the time to establish relationships between classmates is much shorter than the entire semester.

As described in Theorem 1, the network is resilient in the long memory limit, and our model will be separated by the critical value 
ρ1c
 that separates the equal and hierarchical states in the network. There are two aspects to note; first, when the estimate 
φ^
 is very different from the long memory limit, there is no significant hierarchical structure in the network. Second, in each data set, the number of updates *m* of the network uses the average number of updates per step. Using this value and Theorem 1, an approximate critical value 
ρ1c
 can be calculated for the network system in the long memory limit. The next comparison between the data-derived preference estimates of 
ρ1
 and the calculated approximate critical value of 
ρ1c
 reveals the existence of network hierarchy states for all four economic systems (Table 2), with little difference between the estimated 
ρ1
 and the approximate critical value of 
ρ1c
 using RootDegree as the score function. In the economic airline network data with RootDegree as the main model, the estimated value of 
ρ1
 is slightly below the critical value, and conversely, in all three of its datasets, the estimated value is slightly above the critical value, which is more pronounced in the national trade and national investment networks. the RootDegree model has a double steady state (Figure 2a), and in the economic urban airline network, 
ρ1
 estimates are below the critical threshold, but are consistent with the long-term hierarchical structure of the network. Simulations using the inferred parameters as shown in Figure 4 produce a similar long-term hierarchical structure to that observed in the network data. The PageRank model has a similar behavior to the RootDegree model, with a double stable state in the network (Figure 2b). In the SpringRank model, which obtained maximum likelihood values in both the country trade network, country investment network and university economic capacity network, the estimates of 
ρ1
 clearly exceed the system critical threshold and tend to [2, 3]. In conclusion, all three models, SpringRank, PageRank, and RootDegree, show that the system corresponding to each economic network data is in a state of hierarchical structure with continuous individual reinforcement.

Different data have different best-fit scoring functions, while different scoring functions lead to different network rank evolution states. As shown in Figure 4, the original data assignment (Figure 4a) and the RootDegree model (Figure 4d) show strong consistency in the urban economic airline network, with most of the airline routes being in the higher-ranked economic cities. The PageRank and RootDegree models (Figure 5c,d) produce smoother ranking trajectories compared to networks that purely describe rank status because the parameter estimation “memory factor” 
φ
 is relatively large and the support relationships are maintained for a longer period of time. That is, the economic city with more flights is in a higher position in the network, ranking London first in score most of the time. The SpringRank model (Figure 5b) produces a different nature of trajectory, ranking New York first most of the time. This rank variability reflects the sensitivity of the different models to the economic city where the flight is located, and in particular the sensitivity of the SpringRank model to the location of New York, however this is not considered in the other models. In addition, the SpringRank model ranks Tokyo, Paris, and Chicago significantly higher than Atlanta and Beijing, despite the fact that these economic cities have about the same number of flights.

## 4. Conclusions

We propose a mathematically analytic and statistically inferred model of network dynamic evolution to address the problem of shaping and sustaining hierarchical structures among economic networks. When the support for high-ranking nodes in the network exceeds a certain critical value, the equality state relationship is broken and the hierarchical structure emerges. Meanwhile, the transition between equality and hierarchical states in the network depends on the structure of the score function and the preferences of the network for different nodes. The findings suggest that the network evolution generated through transmissive prestige is sufficient to lead to the emergence of hierarchical structures in the network.

Importantly, the likelihood function is introduced in order to allow a good statistical inference of the node preference behavior and memory factors in the network data. In the economic network dataset presented in Section 3, it is clear that a persistent pattern of network rank evolution exists (
ρ1>0
 and 
ρ2<0
), and while inter-network support relationships flow to higher ranked nodes (
ρ1>0
), the real possible associations between nodes are those that are similar to themselves in rank (
ρ2<0
), and such support relationships are not directly associated to the highest level, but rather over time up a few levels.

Our model also has some limitations. For better modeling, we assume the existence of the same preference support parameters for each node in the network, while requiring each node to have knowledge of the network global, which is unlikely in a real system. According to the idea of Li [36,37]. Our model may indicate other avenues for further work. The relationship between time-series networks and correlation centrality measures [29,36] is also of good research interest, as the networks may produce different evolutionary states when different correlation centrality measures are applied to the same economic network dataset. In addition, it is of good research value to predict the important nodes in the risk propagation process of economic networks [28,38]. Extending existing economic network-based models [25], so that their parameters can be statistically learned from the data, while different modeling frameworks can be compared for validation.

## Figures and Tables

**Figure 1 entropy-24-00702-f001:** Schematic diagram of our model dynamics. *t* = 1 initializes the nodes, different colors represent different network nodes, a set of pre-know support matrices are recorded in A (solid arrows) and the node scores *s* = are calculated (vertical axis). Afterwards, the new support relations obtained in the network are added (dashed lines). At the next time period *t* = 2, the old support interactions in the system decay by a factor of 
φ
 (gray arrow). The new support and the decayed old support relationship generate a new score function, which is then executed in the next time period according to this step.

**Figure 2 entropy-24-00702-f002:** Representative dynamics of our proposed model. The population of 
n=8
 nodes is simulated using the SpringRank score function changing at 4000 time steps with 
m=1
 updates per time step, varying the system preference parameters 
ρ1
 and 
ρ2
. Panels (**a**,**c**,**e**,**g**) represent the rank vector 
γ
 simulated over time, different colors tracks the ranks of different nodes. Panels (**b**,**d**,**f**,**h**) represent the adjacency matrix 
A
 at 
t=4000
. Parameters: 
φ
 = 0.995. 
βs
 = 
10−8
.

**Figure 3 entropy-24-00702-f003:** The bifurcation of the SpringRank, PageRank, and Rootdegree score functions model with 
ρ2=0
 and 
m=1
 update per time step. Points give the values of the rank vector 
γ
 averaged over the last 1000 time steps of the 
5×105
-steps simulation with 
n=8
 nodes. The solid line indicates the separation of the nodes into two groups by numerically solving the equation 
f(s,A)=0
, the red curve indicates linear stability and the gray curve indicates linear instability. According to Theorem 1, the vertical line gives the critical value 
ρ1c
. The parameters 
ρ2=0
, 
m=1
, 
φ
 = 0.995, 
βp
 = 
0.85
, 
βs
 = 
10−8
.

**Figure 4 entropy-24-00702-f004:** Simulation of network model dynamics using the parameters 
φ
, 
ρ1
, 
ρ2
 in Table 1. Each row represents one network data, the color traces in the figure represent the top 8 nodes in the network, and the light gray ones represent the other remaining low rank nodes in the network. Other parameters 
βp=0.85
, 
βs
 = 
10−8
.

**Figure 5 entropy-24-00702-f005:** Visualization of the evolutionary ranking function of the economic city aviation network. (**a**): The scores of each economic city, for visualization purposes, are shown as moving averages with a width of 8 years. (**b**): The SpringRank score function is used to infer the rank vector 
γ
 as a function of time. (**c**,**d**): As in (**b**), using PageRank and Rootdegree score functions, respectively. The parameters of panels (**b**–**d**) are shown in Table 1.

**Table 1 entropy-24-00702-t001:** Parameter estimates and likelihood scores for the four datasets described in the main text using SpringRank, PageRank, and RootDegree score functions. The values in parentheses are the standard errors of the parameter estimates (obtained by inverting the Fisher information matrix calculated from the values), the highest log-likelihood 
L
 is indicated in bold, and *N* is the total number of interactions in the network. The trajectory of the inferred parameters is shown in Figure 4.

	SpringRank	PageRank	RootDegree
Ecoaviation(*N* = 1879)	φ^	0.91 (0.01)	0.96 (0.02)	0.87 (0.01)
ρ1^	2.99 (0.03)	0.74 (0.01)	1.28 (0.01)
ρ2^	−1.12 (0.00)	−0.07 (0.02)	−0.18 (0.04)
L	−14,906	−15,106	**−14,308**
IntTrade(*N* = 5524)	φ^	0.67 (0.12)	0.59 (0.09)	0.97 (0.11)
ρ1^	3.03 (0.13)	1.82 (0.06)	0.84 (0.05)
ρ2^	−1.74 (0.11)	−0.5 (0.03)	−0.12 (0.02)
L	**−965**	−1078	−1153
IntInvestment(*N* = 4958)	φ^	0.40 (0.05)	0.13 (0.03)	0.42 (0.06)
ρ1^	2.86 (0.12)	0.82 (0.05)	0.62 (0.03)
ρ2^	−1.46 (0.11)	−0.12 (0.01)	−0.06 (0.02)
L	**−937**	−1036	−958
Friendeco(*N* = 1028)	φ^	0.71 (0.14)	0.81 (0.18)	0.56 (0.15)
ρ1^	2.33 (0.14)	1.21 (0.07)	0.95 (0.05)
ρ2^	−0.86 (0.17)	−0.25 (0.05)	−0.08 (0.02)
L	**−1829**	−1876	−1852

**Table 2 entropy-24-00702-t002:** Comparison of the mean critical values 
ρ1c
 and 
ρ1
 estimates of the system calculated by Theorem 1, with *m* being the average number of interactions at each time step. As in Table 1, the parameters corresponding to the highest log-likelihood are shown in bold, and the upper right * indicates that the estimate is two standard errors more than the critical value, while the lower right 
ρ1^
 indicates that the estimate is two standard errors less than the critical value. The trajectories of the inferred parameters are shown in Figure 4.

	SpringRank	PageRank	RootDegree
Eco aviation	▿	2.01	1.15	**1.35**
ρ1^	2.99 *(0.03)	0.74▿ (0.01)	** 1.28▿ (0.01)**
Int Trade	ρ1c	**2.05**	1.17	0.56
ρ1^	3.03 *(0.13)	1.82 *(0.06)	0.84 *(0.05)
Int Investment	ρ1c	**2.01**	1.19	0.50
ρ1^	2.86 *(0.12)	0.82▿ (0.05)	0.62 *(0.03)
Friend eco	ρ1c	**2.05**	1.18	0.91
ρ1^	2.33 *(0.14)	1.21 *(0.07)	0.95 *(0.05)

## Data Availability

The aviation economy data from the 2016 “Open Airline Airport Database” available at openflights.org (accessed on 10 March 2022) (http://openflights.org/data.html (accessed on 10 March 2022)); International trade data are derived from the Direction of Trade Statistics (DOTS) published by the International Monetary Fund Statistics Yearbook(DOTS); International investment data obtained from the Coordinated Portfolio Investment Survey (CPIS) database provided by the International Monetary Fund; College friend economic data from the KONECT network database.

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
