# Peer review of "RETRACTED: An Adaptive Hierarchical Network Model for Studying the Structure of Economic Network"

_entropy, 2022, doi:10.3390/e24050702_

Round 1

Reviewer 1 Report

The paper addresses the issue of analyzing the structure of some economic networks. The authors are providing a comprehensive literature review on the initial part of the paper along with pointing out their contribution to the field.

I have the following comments regarding the paper:

Please add references to the statement: "The RootDegree score is consistent with previous research findings that the airline economy plays an important role in local production life, transportation, and air transport in the logistics sector. "

Please present more in depth the four datasets you have selected for the experimental results in section 3. I think that the readers are not all familiar with the datasets, so a proper description is needed.

Please compare your results with other works from the field, by applying them on the same dataset.

Please remove Figure 1 as it is not suitable for a scientific paper. In my opinion, can be used for a conference presentation, but in the context of a scientific paper, it brings no value.

Please change "nain" to "main" in the caption of Table 1.

Author Response

Please see the attachment,thank you.

Reviewer 2 Report

In the manuscript entitled "An Adaptive Hierarchical Network Model for Studying the Structure of Economic Network" is presented a hierarchical dynamic model for studying the economic networks.  The manuscript needs a thorough revision and cannot be accepted for publication in Entropy due to the following reasons: 1. In the Introduction the goal of this study should be clearly presented, focusing on the hierarchies existing in economic networks, the dynamics at the root of ranking of economic actors and the stability/instability of actors/relations. The Introduction should be concluded with a clear statement of the model the authors have in mind. 2. The Related Literature should deserve a section by itself. It is fundamental that, instead of a list of papers, a reasoned list of relevant papers is presented. This section should be concluded by the so-called research gap(s), in which the authors discuss the timeliness of their contribution and the improvement with respect to the state of the art they intend to bring forward with their paper. 3. In the section Materials and Methods, some definitions are necessary to allow the reader to plainly follow the discussion. This is the case, for example, of the "recognized support" (page 4, line 114) and the support matrix \omega(t). Other aspects that need to be better introduced are: i) the D matrices (they should express the degree of the nodes of the network), ii) the role of the parameter \beta_s in Eq. 2 and the use of the boldface for s (is it a vector?) and e (it should be defined). Also the choice of the three score functions should be motivated. Why these three functions? Why the RootDegree instead of the degree? Similar considerations apply to Eq. 4 as well. What is the utility model? Why is it expressed by Eq. 4?

Author Response

Please see the attachment,thank you.

Reviewer 3 Report

The author try to apply a hierarchical network model approach to modeling economic and financial networks. While this sounds interesting, much more work is needed:

1) A better motivation and connection with the economics and finance literature on networks.

2) What data do the author use in their estimations?

3) How to interpret the findings in a way that is relevant to the economics and finance literature.

4) A better discussion of results in the light of the key findings in the paper.

Author Response

Please see the attachment,thank you.

Reviewer 4 Report

Overall the study is interesting. I have some comments:

Follow this format the abstract: Introduction > Objective of the study > Methods > Key findings > Implications of the study.

What are the research questions the authors have addressed? What are the objectives of this study? It is not clear from the introduction.

The color codes of Figure 3 are not clear.

The code of gray lines in Figure 6 are not clear.

The practical implications of this study is missing.

Overall the study is interesting. I have some comments:

Follow this format the abstract: Introduction > Objective of the study > Methods > Key findings > Implications of the study.

What are the research questions the authors have addressed? What are the objectives of this study? It is not clear from the introduction.

The color codes of Figure 3 are not clear.

The code of gray lines in Figure 6 are not clear.

The practical implications of this study is missing.

Round 2

Reviewer 2 Report

The revised version of the manuscript entitled "An Adaptive Hierarchical Network Model for Studying the Structure of Economic Network" contains a relevant improvement with respect to the first version. All my concerns and suggestions have been satisfactorily addressed and I am glad to recommend it for publication. It is important, however, that an extension revision be carried out on the Bibliography: for example at page 4 Caterina De Bacco et al is erroneously referenced to as Caterina et al. 

Reviewer 3 Report

The authors have addressed all issues raised in my the referee report.

Reviewer 4 Report

Thank you for the corrections.